# Mapping Bone Marrow Cell Response from Senile Female Rats on Ca-P-Doped Titanium Coating

**DOI:** 10.3390/ma15031094

**Published:** 2022-01-30

**Authors:** Leonardo P. Faverani, William P. P. Silva, Cecília Alves de Sousa, Gileade Freitas, Ana Paula F. Bassi, Jamil A. Shibli, Valentim A. R. Barão, Adalberto L. Rosa, Cortino Sukotjo, Wirley G. Assunção

**Affiliations:** 1Department of Diagnosis and Surgery, Division of Oral and Maxillofacial Surgery and Implantology, School of Dentistry, São Paulo State University—UNESP, Araçatuba 16015-050, Brazil; leonardo.faverani@unesp.br (L.P.F.); william.phillip@unesp.br (W.P.P.S.); ana.bassi@unesp.br (A.P.F.B.); 2Dental Prosthesis, Department of Dental Materials and Prosthodontics, School of Dentistry, São Paulo State University—UNESP, Araçatuba 16015-050, Brazil; cecilia.sousa@unesp.br (C.A.d.S.); wirley.assuncao@unesp.br (W.G.A.); 3Bone Res Lab, School of Dentistry of Ribeirao Preto, University of São Paulo (USP), Ribeirão Preto 14040-904, Brazil; gileadefreitas@usp.br (G.F.); adalrosa@forp.usp.br (A.L.R.); 4Department of Periodontology and Oral Implantology, Dental Research Division, University of Guarulhos (UnG), Guarulhos 07023-070, Brazil; jshibli@ung.br; 5Department of Prosthodontics and Periodontology, Piracicaba Dental School, University of Campinas (UNICAMP), Piracicaba 13414-903, Brazil; vbarao@unicamp.br; 6Department of Restorative Dentistry, College of Dentistry, University of Illinois at Chicago, Chicago, IL 60612, USA

**Keywords:** oxidation, stromal mesenchymal cells, dental implants, titanium surfaces

## Abstract

Chemical and topographical surface modifications on dental implants aim to increase the bone surface contact area of the implant and improve osseointegration. This study analyzed the cellular response of undifferentiated mesenchymal stem cells (MSC), derived from senile rats’ femoral bone marrow, when cultured on a bioactive coating (by plasma electrolytic oxidation, PEO, with Ca^2+^ and P^5+^ ions), a sandblasting followed by acid-etching (SLA) surface, and a machined surface (MSU). A total of 102 Ti-6Al-4V discs were divided into three groups (n = 34). The surface chemistry was analyzed by energy dispersive spectroscopy (EDS). Cell viability assay, gene expression of osteoblastic markers, and mineralized matrix formation were investigated. The cell growth and viability results were higher for PEO vs. MSU surface (*p* = 0.001). An increase in cell proliferation from 3 to 7 days (*p* < 0.05) and from 7 to 10 days (*p* < 0.05) was noted for PEO and SLA surfaces. Gene expression for OSX, ALP, BSP, and OPN showed a statistical significance (*p* = 0.001) among groups. In addition, the PEO surface showed a higher mineralized matrix bone formation (*p* = 0.003). In conclusion, MSC from senile female rats cultured on SLA and PEO surfaces showed similar cellular responses and should be considered for future clinical investigations.

## 1. Introduction

With the increase in life expectancy, there is a greater need for orthopedic and dental rehabilitation due to pathological or traumatic fractures, joint wear, or the loss of dentition. Osseointegration of biomaterials depends not only on the properties of the implanted biomaterial but also on the patient’s characteristics and the osteogenic capacity [1].

A significant challenge in this rehabilitative process is the decrease in the quantity and quality of bone tissue caused by osteoporosis [2]. Low-density bones routinely found in osteoporotic, diabetes, decompensated hypertensive patients, etc., represent a significant challenge in successful implants osseointegration [2,3,4,5]. Due to senility, primary osteoporosis has become an important health concern worldwide, as it is an age-related disorder (over 70 years) and causes changes in cortical and trabecular bone. In addition to the imbalance in bone formation and resorption, evidence shows changes in the quantity and function of bone marrow mesenchymal stem cells in senility [6].

To promote and increase the bone’s surface contact area with the implant, chemical and topographical modifications on the implant surfaces have been widely studied to improve the biological responses of the early immune-inflammatory process, angiogenesis, and osteogenesis [7]. Implant surface topography can facilitate cell migration and proliferation during bone repair. In addition, boosting the osteoblastic differentiation of mesenchymal stem cells optimizes and accelerates the implant osseointegration process [8].

Surface modifications based on a subtractive method, such as sandblasting and acid etching, has been widely studied [9]. The use of chemicals to modify the material’s surface generates hydroxyl (-OH) groups on the oxide layer [10,11,12,13], leading to a better interplay between proteins and atoms. In this regard, SLA is one of the most common implant surfaces used by many clinicians. It is produced by blasted corundum particles followed by an acid etching procedure; Straumann AG—Basel, Switzerland) [9,14]. Several studies have proven the effectiveness of these surfaces since they promote more hydrophilic surfaces, contributing to the acceleration and increase in bone apposition [9] by adsorbing proteins that affect cell adhesion’s early regulation. In addition to stabilizing blood clots and early vascularization at the repair site, it can play an essential role during the early stages of peri-implant wound healing [14,15], shortening the healing period [14,16].

Biomaterials need to have osteoinduction, osteoconduction, osteopromotion properties, as well as long-term stability and biocompatibility. Numerous studies describe favorable bioactivity on porous surfaces [7,8,9,14,15,17,18,19,20,21,22,23]. Roughness and porosity are two interdependent surface characteristics of bioactive surfaces that result from the manufacturing process. Modification with ion incorporation on dental implant surfaces allows for the fabrication of biocompatible and bioactive titanium surfaces that exhibit improved behavior concerning attachment and cell growth compared to conventional titanium surfaces [1,2,6,20].

Thus, the possibility to design bioactive coatings on implant surfaces is gaining great attention. Among them, a simple and straightforward method is the electrolytic plasma oxidation (PEO). It is defined as an advanced electrochemical technique based on anodization at high voltages exceeding the dielectric breakdown voltage of the oxide layer on the metal surface and the gas envelope [9,14,15,16,17,22,23,24,25,26,27]. Based on microdischarges, it allows for the formation of a coating with high adhesive strength and micropores, with the ability to incorporate ions (Ca^2+^, P^5+^, Zn^2+^, Mg^2+^, Ag^1+^, Sr^2+^, Si^2+^) [9,14,15,16,17,20,21,22,23,26,27]. As a result, numerous plasma microdischarges are generated that facilitate rapid coating growth and the formation of new oxide phases for bioactivation [17]. Thus, this technology is particularly relevant because of the coating formed ensures higher wear and corrosion resistance, higher protein absorption [15,17], osseointegration, and can encompass drug delivery system [9,15,17]. However, no consolidated information can be found on the interplay between the PEO surface and bone cell behavior in a low bone quality condition.

The main objective of dental implants research has been to obtain a surface treatment capable of assisting or even improving osseointegration and decreasing rehabilitation failure. Besides allowing a short time of treatment, such a surface may be capable of bringing more osteogenic cells faster, especially in low bone quality situations, such as osteoporosis, diabetes, systemic arterial hypertension, hypovitaminosis D, hypocalcemia, etc. 

For implants companies to start manufacturing a new biomaterial or a surface treatment, it is necessary to gather data to provide scientific evidence. One of the critical aspects to be assessed is the peri-implant bone response, especially in low bone mineral density conditions since it is still a challenge for osseointegration. Preclinical tests through animal experiments design represent a good choice; however, it is difficult to obtain enough samples for all of the analyses intended. This is mainly due to some loss of animals as the old age and the development of other systemic decompensation after surgeries procedures. Therefore, in vitro tests using primary osteogenic cells are more appropriate and provide consistent results using few animals.

Given the search for other methods of surface treatment for implants and the improvement in osseointegration responses, this study aimed to analyze the influence of PEO-treated surface with the addition of Ca^2+^ and P^5+^ ions on Ti-6Al-4V alloy discs, compared to a widely used therapy by combining double acid etching with aluminum blasting (SLA surface) on the interaction with undifferentiated mesenchymal stem cells from the femoral bone marrow of senile rats. The null hypothesis was that there would be no significant difference in osteogenic cell responses between the two surfaces investigated in this study (PEO versus SLA).

## 2. Materials and Methods

### 2.1. Specimens

A total of 102 Ti-6Al-4V discs with 10 mm diameter and 2 mm thickness were supplied by Emfils Dental Implants (Itu, São Paulo, Brazil). Discs were divided into three groups (n = 34 per group): (1) machined surface (MSU) group (n = 34); (2) SLA group: surface texturing according to the company’s standards (nitric acid, neutral detergent, 95% alcohol, drying, aluminum oxide blasting, 99% alcohol, nitric acid, neutral detergent, distilled water, 95% alcohol, drying, and packaging); (3) PEO group: surface treatment with the addition of Ca^2+^ and P^5+^ ions.

### 2.2. Plasma Electrolytic Oxidation Treatment

PEO coating was conducted through an electrical system and a reactor consisting of an electrode holder and electrolytic tank. The electrolyte solution was prepared by dissolving Ca(NO_3_)_2_ 4H_2_O, NH_4_H_2_PO_4_ (3.6 × 10^4^ M) in 1 L of distilled water, with a molar ratio of 1.67 [16,25]. Then, using the electrical system (variable output AC power supply, transformer, a rectifier circuit, circuit breaker, ammeter, and voltmeter), the electrodes were fed with a DC voltage up to 1000 V and a maximum current of 1.5 A. An AC voltage inverter coupled to the source allowed the adjustment for the voltage value. The oxidation parameters were current density of 50 mA/cm^2^ and voltage of 290 V over 10 min, with the temperature maintained at 15 ± 2 °C.

### 2.3. Chemical Characterization

The elemental chemical composition (in atomic% and in the order of 1 μm^3^) of the surfaces was screened using the energy-dispersive spectroscopy (EDS) (VANTAGE Digital Microanalysis System, Noran Instruments Inc., Middleton, WI, USA) coupled to the scanning electron microscope (SEM). To guarantee reproducibility, three random areas were selected in each micrograph (×2000, 10 kV, WD = 11 mm) [28].

### 2.4. Animals

The study was submitted and approved by the Local Animal Ethics Committee of the Araçatuba School of Dentistry, Araçatuba, Brazil (FOA-UNESP; # 01040-2016). Three Wistar rats (Rattus Norvergicus), aged 18 months, weighing 350 to 400 g, were selected for this study. Throughout the experiment, the animals were kept in cages under a stable temperature environment (22 ± 2 °C), controlled light cycle (12 h of light and 12 h of darkness), and fed with solid food (Presence^®^, Paulínia, São Paulo, Brazil).

### 2.5. Isolation and Cell Culture

Animals were euthanized by anesthetic overdose (thiopental sodium 150 mg/kg body weight), intraperitoneal lidocaine, followed by skin antisepsis in the femoral region. The femurs were removed and transported in a culture medium containing alpha-minimal essential medium (α-MEM) (Gibco-Life Technologies, Waltham, MA, USA) with the supplementation of 500 µg/mL gentamicin and 3 µg/mL fungizone (Gibco-Life Technologies, Waltham, MA, USA) [29,30,31]. Femoral bone marrows were extracted inside the laminar flow chamber by irrigation of growth media (non-differentiating condition) containing α-MEM supplemented with 10% fetal bovine serum (Gibco-Life Technologies, Waltham, MS, USA), with an addition of 50 μg/mL gentamicin (Gibco-Life Technologies), and addition of 0.3 μg/mL fungizone (Gibco-Life Technologies, Waltham, MA, USA).

Bone marrow mesenchymal stem cells (MSC-BM) were cultivated in α-MEM supplemented with 10% fetal bovine serum for 7 days, and the culture medium was changed every 72 h under controlled conditions (37 °C and 5% CO_2_).

### 2.6. Cell Plating on Ti Discs

After obtaining an adequate number of MSC-BM, the cells were trypsinized from the cell culture flask and plated directly onto sterile Ti discs (MSU, SLA, and PEO groups) with 1.8 mL modified α-MEM (360 mL α-MEM, 40 mL fetal bovine serum (Gibco-Life Technologies, Waltham, MA, USA), 2 mL gentamicin (Gibco-Life Technologies, Waltham, MA, USA), 500 μL fungizone (Gibco-Life Technologies, Waltham, MA, USA), 4 mL dexamethasone (Sigma-Aldrich, San Luis, MO, USA), 4 mL β-glycerophosphate (Sigma-Aldrich, San Luis, MO, USA) and ascorbic acid (Gibco-Life Technologies, Waltham, MA, USA). As a positive control of the experiment, cells were placed directly on the plastic culture plate without Ti discs.

### 2.7. Cell Growth and Viability

To evaluate cell growth and viability, cells were cultured (2 × 10^4^ cells/well) on MSU, SLA, and PEO discs (4 discs/group) for 3, 7, and 10 days. MTT (3[4,5-dimethylthiazol-2-yl]-2,5-diphenyltetrazolium) assay (MTT, Sigma-Aldrich) was used to measure cell growth and viability. First, cells were incubated at 37 °C for 4 h in 100 mL of MTT (5 mg/mL PBS). After the incubation period, 1 mL of acidic isopropanol (0.04 N HCl in isopropanol) was added to each well and subjected to stirring for 5 min. Finally, 150 mL of the solution was added in 96-well plates and analyzed by optical density, considering 570 nm as a reference in a Quant spectrophotometer (Biotek, Winooski, VT, USA) (n = 5).

### 2.8. Real-Time qPCR Analysis

For real-time PCR analysis, cells (2 × 10^4^ cells/well) were cultured on 12 Ti discs from each group (MSU, SLA, and PEO). On day 7, total RNA was extracted using the Total RNA Isolation System kit (Promega, Madison, WI, USA) assessing the relative gene expression of runt-related transcription factor 2 (RUNX 2), Osterix (OSX), bone sialoprotein (BSP), and osteopontin (OPN). The Trizol reagent (Life Technologies-Invitrogen, Carlsbad, CA, USA) was used to extract the total RNA, and to be continued by SV total RNA isolation system (Promega, Madison, WI, USA) and quantified from 1 µL of the sample (NanoVue-GE Healthcare, USA). Its integrity was assessed by 1.5% agarose gel electrophoresis. The preparation of a cDNA strand was performed from 1 µg of total RNA (Mastercycler Gradient—Eppendorf, Hamburg, Germany) by reaction with the reverse transcriptase enzyme (SuperScript^™^ III First-Strand Synthesis Systems kit, Invitrogen-Life Technologies, Carlsbad, CA, USA).

Using the TaqMan probe system (Invitrogen-Life Technologies, Carlsbad, CA, USA) on the 7500 Fast Real-Time PCR System (Applied Biosystems, Waltham, MA, USA), real-time PCR reactions were induced. PCR reactions were performed in triplicate with 10 µL final volume containing 5 µL of TaqMan Universal PCR Master Mix-No AmpErase UNG (2×), 0.5 µL of TaqMan probes for the genes of interest (20× TaqMan Gene Expression Assay Mix), and 11.25 ng/µL cDNA. The relative gene expression results were normalized by the constitutive gene β-Actin and calibrated by the control group (n = 3).

### 2.9. Alkaline Phosphatase Activity

On the tenth day, an analysis of the alkaline phosphatase activity was performed. The cells were plated on 4 Ti discs from each group (MSU, SLA, and PEO) at the concentration of 2 × 10^4^ cells/well. First, 50 µL of thymolphthalein monophosphate was mixed with 0.5 mL of 0.3 M diethanolamine buffer, pH 10.1, during 2 min at a temperature of 37 °C. It was added to 50 µL of the fragments obtained from each cell for 10 min at 37 °C (Labtest Diagnóstica, Lagoa Santa, MG, Brazil), and then, 2 mL of 0.09 M Na_2_CO_3_ and 0.25 M NaOH were added for a period of 30 min. The measurement of absorbance was performed at 590 nm. The standard curve with thymolphthalein (range of 0.012–0.4 mmol in h/mL) was used to measure the ALP activity. The data obtained were expressed as the ALP activity normalized to the total protein content [32].

### 2.10. Analysis of Mineralized Matrix Formation

At 21 days, cultured cells from each disc were fixed with 10% buffered formalin for 24 h at 4 °C (n = 5). After this period, formalin was removed, dehydration was performed using a graded series of alcohols, and the discs were stained with 2% alizarin red (Sigma) (pH 4.2) for 10 min at room temperature. For qualitative analysis, the discs were photographed by a high-resolution digital camera (Canon EOS Digital Rebel Camera, Canon, Lake Success, NY, USA). Measurement of the calcium content was performed using 150 µL of 10% acetic acid placed in each well under gentle agitation for 30 min. The cell layer was scraped off the disc’s surface and added to the solution. Then, it was vortexed for 30 s, heated to 85 °C for 10 min to break the nodules, and cooled on ice for 5 min. For 15 min, the solution was centrifuged (20,000× *g*), and the supernatant was transferred, along with the addition of 40 µL of 10% ammonium hydroxide. The results were analyzed by optical density considering absorbance at 405 nm (μQuant, Bio-Tek Instruments Inc., Winooski, VT, USA).

To summarize, the experimental design of this study is schematized in Figure 1.

### 2.11. Statistical Analysis

After the normality test (Shapiro–Wilk), the cell viability results were analyzed using two-way ANOVA test and the Holm–Sidak method for multiple comparisons when necessary. Data obtained by real-time PCR, alkaline phosphatase activity, and mineralization nodules were analyzed by one-way ANOVA and Kruskal-Wallis test. When necessary, the Tukey test was used as a post-test. The significance level of 5% was considered for all analysis in the SigmaPlot 12.0 statistical program (Exakt Graphs and Data Analysis, San Jose, CA, USA).

## 3. Results

### 3.1. Chemical Characterization

The mapping of the elemental chemical composition of the PEO-treated surface confirms the incorporation of Ca^2+^ and P^5+^ in addition to the chemical elements present in the alloy material (Ti^2+^, Al^2+,^ and V^2+^) (Figure 2).

### 3.2. Cell Growth and Viability

Figure 3 illustrates cell viability on MSU, SLA, and PEO surfaces at 3, 7, and 10 days of culture. In general, the PEO surface exhibited lower MSC-BM cell viability compared to SLA and MSU surfaces for all time points (*p* < 0.05) The SLA surface did not differ from MSU group for all time points (*p* = 0.437). Considering the different time points, there was an increase in cell proliferation from 3 to 7 days (*p* = 0.01) and from 7 to 10 days (*p* < 0.05) for PEO surface. Similarly, there was an increase in cell proliferation from 3 to 7 days and from 7 to 10 days (*p* < 0.05) for the SLA surface.

## 4. Gene Expression

The gene expression for all surfaces can be assessed in Figure 4. The SLA surface upregulates the expression of OSX, BSP, and OPN compared to the other groups (*p* = 0.001). However, RUNX2 gene expression was similar among groups (*p* = 0.307).

### 4.1. Alkaline Phosphatase Activity

The alkaline phosphatase activity on the 10th day of culture was similar among SLA, PEO, and MSU (*p* = 0.082) (Figure 5).

### 4.2. Mineralized Matrix Formation

The PEO-treated surface showed a higher mineralized bone formation (*p* = 0.003) in terms of mineralized matrix formation compared to SLA and MSU surfaces (Figure 6). Greater mineral nodules can be visualized in the PEO surface.

## 5. Discussion

The modified surface topographies of the Ti discs used in this study (SLA and PEO) showed favorable results [27] in the cellular response of mesenchymal stem cells derived from senile rats. The differentiation of MSC-BM into osteoblasts has well-defined steps in the literature [33], both in healthy cells and in osteoporosis-induced cells [1,34], as well as the correlation of the influence of bioactivity on implant surfaces in the osseointegration process and cell differentiation [35,36,37,38,39].

The cell differentiation and mineralization of senile MSC-BM obtained in this study reinforce the influence of this bioactivity, demonstrating that the topographical characteristics of the PEO-treated surface favor good results, even in cells in an osteoporotic process. When compared with the topographical features and bioactivity of the SLA-treated surface, which is well documented and widespread, both in the literature and commercially [10,27,40,41,42,43,44], similar and even superior results in cellular responses were noted.

Both modified surfaces (SLA and PEO) showed an increase in cell viability over time. However, the SLA surface outnumbered the PEO surface in all time points (3, 7, and 10 days). This correlates with the literature regarding the culture and differentiation of osteoporotic or normal density cells and clinical studies evaluating osseointegration comparing various surface treatments [1,11,34,36,39,44,45]. It is noted that the PEO surface has a beneficial influence on proliferation, being more accelerated, in the process of differentiation into osteoblasts, while the SLA surface is still in the process of cell proliferation [42].

The formation of regular pores on the PEO-treated surface appears to facilitate cell adaptation and adhesion Conserva (2013) and Plekhova (2020), but the higher irregularity of the SLA surface makes cell adaptation more challenging. These findings are likely to account for the better bone response observed in the current investigation for PEO disc surfaces. However, in statistical terms, it was apparent that both surfaces allowed for MSC-BM differentiation [46,47].

When the gene expression at day 7 was evaluated, indicating the initial phases of differentiation of MSC-BM into osteoblasts through the expression of genes RUNX2 and OSX, and in an “intermediate” phase characterized by the beginning of apatite precipitation, represented by the gene expression of BSP, higher values were noted for the SLA surface, denoting that the PEO surface presents in a more advanced stage. The results of gene expression of OPN, being considered as a “final” phase in the differentiation of MSC-BM, which already present mineralization, corroborate with the data previously described [42,48], in which the PEO surface reached bone maturation faster than SLA; this followed the regular chronological phases of bone tissue repair.

This fact agrees with the literature and may benefit the difficulties found in clinical situations of regeneration/osseointegration process in patients with a decrease in bone quality, such as diabetic patients, osteoporotic patients, etc. [1,4,5,30,49,50,51,52,53,54,55,56]. Thus, it is evident that when the healing process occurs in a shorter time, bone remodeling benefits in the cases cited above. Thus, the PEO-modified surface presents a similar and even superior approach to implant healing compared to the traditional surfaces already found in the market (SLA).

As the cell differentiation and mineralization process progress, an increase in alkaline phosphatase activity occurs, with similar results when comparing the tested groups (SLA and PEO) at day 10. However, on the twenty-first day, when analyzing the surfaces’ mineralization nodules, the PEO surface shows superiority. The greatest mineralization nodules formation on the PEO surface is in line with studies by Kazek-Kesik et al. (2015), who reported the facilitation in collagen production and cell activity mineralization due to the surface characteristics promoted by PEO [24,57].

This benefit becomes even more evident when correlated to senile conditions, where the presence of deficient bone is even more apparent [6]. This is evidenced by the studies of Polo et al. (2020) and Momesso et al. (2020), where the PEO coating was investigated on implants installed in rat tibiae after induction of osteoporosis by bilateral ovariectomy compared to the acid-blasted implant surface (SLA). At the end of the analysis, there was no difference between the two coatings. However, more significant peri-implant bone formation was noted at the interface between the remaining bone of osteoporotic rats and the PEO-coated implant surface than SLA [42,48]. Furthermore, corroborating our study results, the immunohistochemical reactions and PCR assay showed mineralization proteins being expressed early and decreasing toward the end for the PEO group, confirming once again that the anodizing coating can increase bone response [42,48].

The choice of primary osteogenic cells from senile bone marrow rats in this study was to create in vitro a critical bone condition that the bone/implant interface integration is more vulnerable to [2,3,4,5]. Developing a surface morphology that accelerates bone healing is one of the top interests in materials sciences and biotechnology. The high predictability and good results obtained in daily clinical practice through SLA surface treatment on Ti and Ti alloy implants, highlight this texturing method as one of the main techniques used in the market [11,58,59,60,61,62].

Marques et al. (2015) demonstrated in their study that the PEO textured surface with the incorporation of calcium ions, phosphorus, silver, and silica, in different concentrations, showed a more homogeneous surface structure, with large pores, an antibacterial property, a greater resistance to corrosion and tribocorrosion, and better mechanical strength [16,25]. In vivo studies comparing the biological behavior of the PEO surface with calcium and phosphorus ions revealed better osseointegration in the final stages of mineralization than other surfaces (SLA) [42,48]. In light of these previous findings, which are supported by this research, it is even more important to conduct a clinical trial to assess the long-term behavior of biomedical implants in patients.

This study presented some limitations, and two are more relevant in clinical speculation. The first one was the bone response under discs design. In the clinical behavior, dental implants obtain threads and need to be placed similarly to a “screw in a wall” to allow peri-implant bone healing during the osseointegration. Another point directly related to the first aspect is the experiment design. A prospective clinical study showed a better design to assess the proposal tested here. Although it is challenging to perform a study with a low-bone density population, the authors believe that future research should investigate implants with PEO surfaces compared to other commercial surfaces. Despite that, in vitro studies are fundamental before manufacturing, which shows this important step to the biomedical literature.

## 6. Conclusions

Cell behavior on the different textured surfaces tested (PEO and SLA) showed similar cell differentiation, maturation, and mineralization responses. Therefore, future clinical prospective studies should consider assessing PEO and other commercially available surfaces.

## Figures and Tables

**Figure 1 materials-15-01094-f001:**
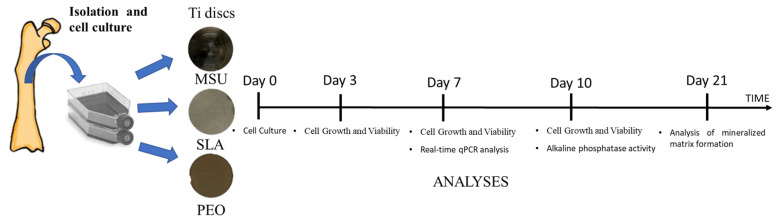
Schematic representation of the experimental groups and periods of the analyses.

**Figure 2 materials-15-01094-f002:**
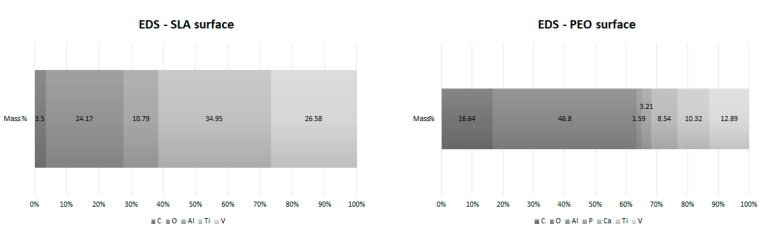
Energy dispersive spectroscopy of Ti6Al4V alloy (percentage by weight—wt%) for SLA surface and PEO surface.

**Figure 3 materials-15-01094-f003:**
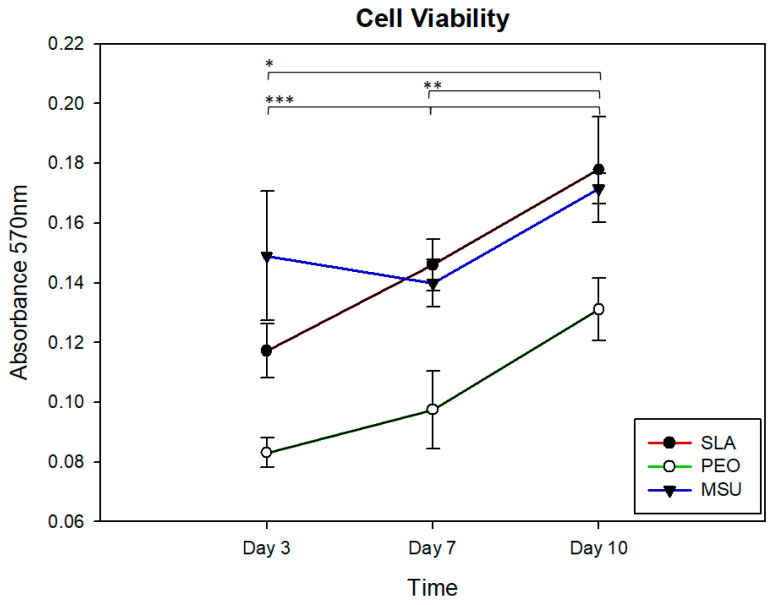
Cell growth and viability for SLA surface and PEO-treated surface under cell culture conditions at days 3, 7, and 10. SLA surface—* 3 to 7 days and from ** 7 to 10 days (*p* < 0.05); PEO surface—*** 3 to 7 days (*p* = 0.01) and from 7 to 10 days (*p* < 0.05).

**Figure 4 materials-15-01094-f004:**
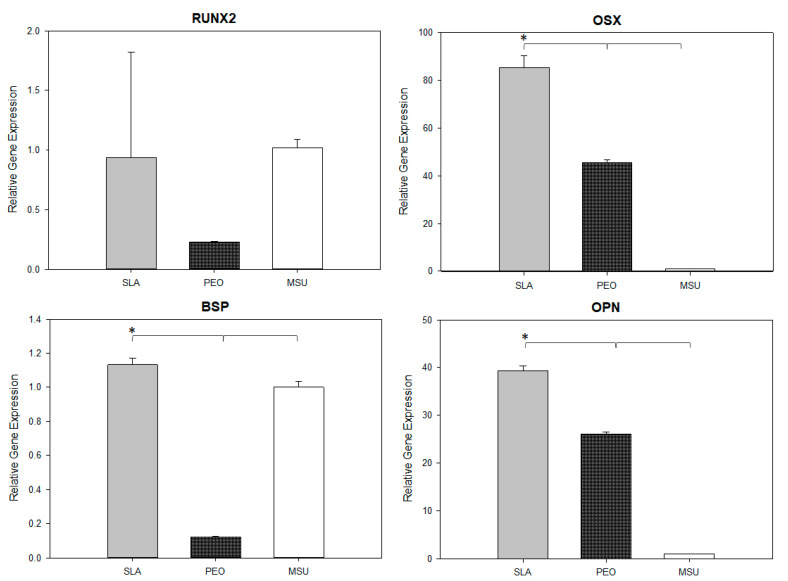
Real-time PCR analysis to determine the expression of RUNX2, OSX, ALP, BSP, OC, and OPN for the SLA surface and PEO-treated surface under cell culture conditions. * represents a significant difference between surfaces (*p* < 0.05).

**Figure 5 materials-15-01094-f005:**
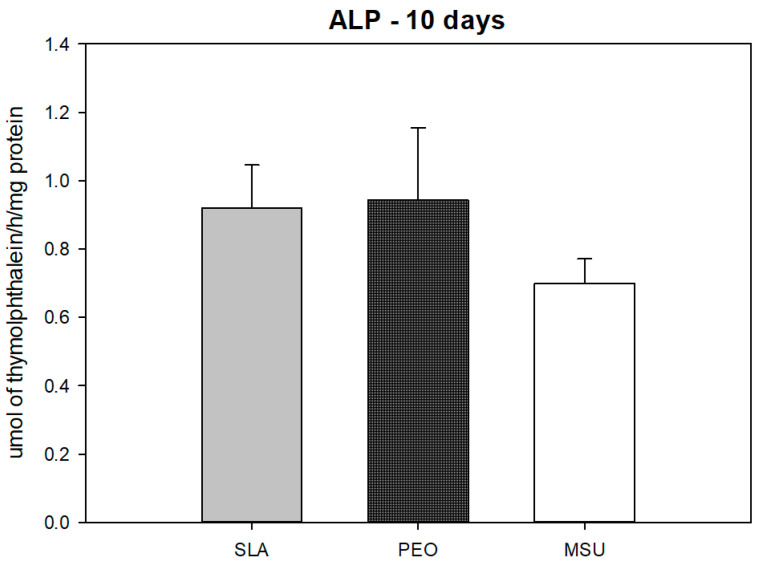
Alkaline phosphatase activity for SLA, PEO and MSU surfaces (*p* = 0.859), showing a similarity in the results.

**Figure 6 materials-15-01094-f006:**
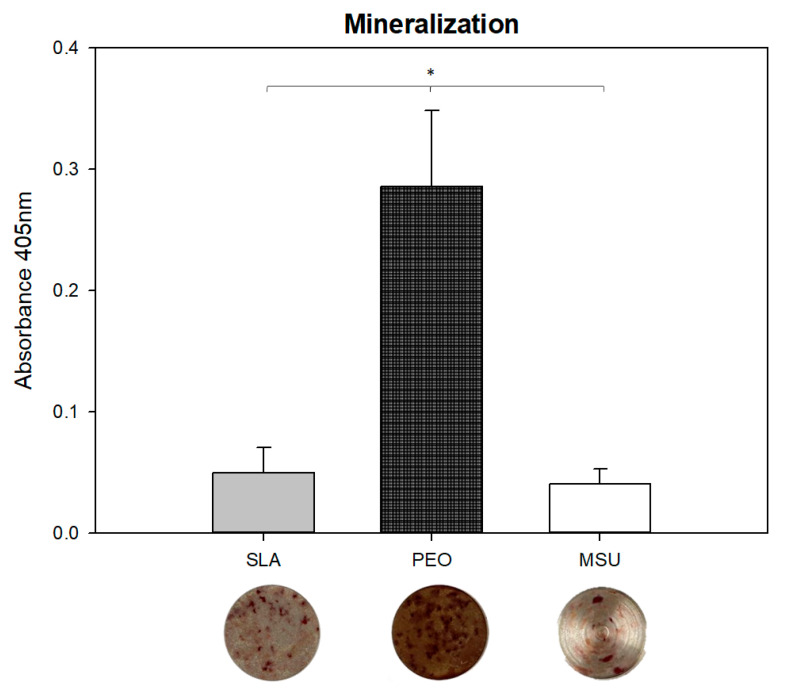
Mineralized matrix formation in the SLA, PEO and MSU surfaces. * represents a significant difference between surfaces (*p* < 0.05). The bottom images show the mineralized nodule formation on each surface.

## Data Availability

The data presented in this study are available on request from the corresponding author. The data are not publicly available due to be part of the patent proposal.

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
