# Peer review of "Mapping Bone Marrow Cell Response from Senile Female Rats on Ca-P-Doped Titanium Coating"

_materials, 2022, doi:10.3390/ma15031094_

Round 1

Reviewer 1 Report

The paper is of interest and well written, however it would benefit of some revision as indicated below.

Keywords: add titanium surfaces

Introduction: add the null hypothesis tested

Materials and Methods, first sentence of paragraph 2.3: please rephrase, the syntax is not correct.

Discussion

The Discussion section should be improved comparing the outcomes with those of other similar studies. In particular:

  • How do the cell type might have affected the outcomes? Are there other published studies dealing with dental implants surfaces using the same type of cells?
  • How do the authors’ outcomes compare with those of other in vitro studies evaluating osteoblasts gene expression of next to the same implant surfaces?

i.e. Similarly to the outcomes herein presented, the study by Baldi et al. (Baldi D, Longobardi MG, Cartiglia C, La Maestra S, Pulliero A, Bonica P, Micale R, Menini M, Pera P, Izzotti A. Dental Implants Osteogenic Properties Evaluated by cDNA Microarrays. Implant Dentistry 2011;20:299-304) did not find any difference in cell proliferation and viability between the different dental implants tested, although Straumann SLActive showed the best results. Conversely, gene expression was revealed to be a more sensitive biomarker being remarkably different in cells adherent to different implants. The 5 dental implants tested significantly modulated the expression of 14 osteogenic activities mainly including bone morphogenetic proteins, osteomodulin, and osteoprotegerin.

To implement the discussion section also the paper by Conserva et al. might be helpful (Conserva E, Menini M, Ravera G, Pera P. The role of surface implant treatments on the biological behavior of SaOS-2 osteoblast-like cells. An in vitro comparative study. Clinical Oral Implants Research 2013;24:880-889).  

The authors should acknowledge the limits of the present in vitro research (i.e. the use of titanium disks instead of dental implants manufactured for clinical use) and discuss how the outcomes of the present in vitro research might be transferred to clinical practice.

Author Response

The authors wish to thank the reviewers for the detailed comments they held on our manuscript. Please find enclosed the point-by-point responses to the Reviewers’ comments. The changes made in the text were highlighted in yellow color for Reviewer 1, and green color for Reviewer 2. 

# Reviewer 1:

1) Keywords: add titanium surfaces

Response: Thank you for your suggestion. This keyword was added.

2) Introduction: add the null hypothesis tested

Response: It was done (please see page 5, paragraph 3).

3) Materials and Methods, first sentence of paragraph 2.3: please rephrase, the syntax is not correct.

Response: Our apologies. The phrase was re-written (highlighted in yellow color).

Discussion

4) The Discussion section should be improved comparing the outcomes with those of other similar studies. In particular: How do the cell type might have affected the outcomes? Are there other published studies dealing with dental implants surfaces using the same type of cells?

Response: Thank you for your suggestion. According to the literature search, no similar studies were found using primary mesenchymal stem cells from senile rats and no article using the same surface treatment proposed (PEO with the association of Ca and P). It is important to have in mind that there are some published manuscripts investigating PEO surface; however, our proposal is to use this technique and incorporate a specific amount of Ca and P onto Ti disk surface to resemble the hydroxyapatite content of the bone tissue. Previous studies have tested those concentrations (Marques et al. 2015):

Marques, I.D.; Barao, V.A.; da Cruz, N.C.; Yuan, J.C.; Mesquita, M.F.; Ricomini-Filho, A.P.; Sukotjo, C.; Mathew, M.T. Electrochemical behavior of bioactive coatings on cp-Ti surface for dental application. 2015, 100, 133-146, doi:10.1016/j.corsci.2015.07.019).

Marques Ida S, d.C.N., Landers R, Yuan JC, Mesquita MF, Sukotjo C, Mathew MT, Barão VA. Incorporation of Ca, P, and Si on bioactive coatings produced by plasma electrolytic oxidation: The role of electrolyte concentration and treatment duration. Biointerphases 2015, 10, 041002.

 For the differential of the study, we highlight the importance of the positive results for both tested surfaces, in the capacity the MSC-BM differentiation and bone formation. Since the MSC-BM cells used originated from senile rats, their capacity for differentiation and bone formation is reduced when compared to cells from young and healthy rats (Fini, M.; Giavaresi, G.; Torricelli, P.; Borsari, V.; Giardino, R.; Nicolini, A.; Carpi, A. Osteoporosis and biomaterial osteointegration. Biomed Pharmacother 2004, 58, 487-493, doi:10.1016/j.biopha.2004.08.016). (Manolagas, S.C. Cellular and molecular mechanisms of osteoporosis. Aging (Milano) 1998, 10, 182-190, doi:10.1007/BF03339652).

 Aiming at obtaining answers about the cellular behavior, mimicking one of the greatest dental clinical challenges which is the rehabilitation treatment in osteoporotic patients or patients with low-bone density due to systemic alterations, we decide to use primary cells from senile rats, which might be consider a novelty in our study.

5) How do the authors’ outcomes compare with those of other in vitro studies evaluating osteoblasts gene expression of next to the same implant surfaces?

i.e. Similarly to the outcomes herein presented, the study by Baldi et al. (Baldi D, Longobardi MG, Cartiglia C, La Maestra S, Pulliero A, Bonica P, Micale R, Menini M, Pera P, Izzotti A. Dental Implants Osteogenic Properties Evaluated by cDNA Microarrays. Implant Dentistry 2011;20:299-304) did not find any difference in cell proliferation and viability between the different dental implants tested, although Straumann SLActive showed the best results. Conversely, gene expression was revealed to be a more sensitive biomarker being remarkably different in cells adherent to different implants. The 5 dental implants tested significantly modulated the expression of 14 osteogenic activities mainly including bone morphogenetic proteins, osteomodulin, and osteoprotegerin.

Response: The comparison of our study with the suggested study is a little difficult, due to the methodological differences related to cell origin and the proposed surface treatment (PEO with the addition of Ca and P). We emphasize the absence of studies in the literature with the same methodology. The purpose of our study was to obtain results demonstrating the influence of surface treatments on the differentiation capacity of MSC-BM and bone formation, from senile female rats, showing at least an osteopenic condition. Similar studies, as the study suggested, present results from osteoblastic cell lines (SAOS-2 Osteoblasts), which differ significantly from a mesenchymal stem cell in senility. Even with the limitation of our study, it was possible to notice that the influence of surface treatments helps in cell differentiation, formation, and bone mineralization of physiologically weakened cells.

6) To implement the discussion section also the paper by Conserva et al. might be helpful (Conserva E, Menini M, Ravera G, Pera P. The role of surface implant treatments on the biological behavior of SaOS-2 osteoblast-like cells. An in vitro comparative study. Clinical Oral Implants Research 2013;24:880-889).

Response: Thank you for your consideration, and we have added the reference in the discussion of our paper, which you can find on page 16, paragraph 03.

7) The authors should acknowledge the limits of the present in vitro research (i.e. the use of titanium disks instead of dental implants manufactured for clinical use) and discuss how the outcomes of the present in vitro research might be transferred to clinical practice.

Response: The limitations of the study and future proposals are essential for research proposals. A paragraph discussing these aspects was added on page 17; third paragraph; highlighted in yellow color. Thank you.

Reviewer 2 Report

Based on the results of an in-depth evaluation that I have done for a research article with the title "Mapping bone marrow cell response from senile female rats on 1 Ca-P-doped titanium coating", I think this manuscript should be rejected because it is not appropriate for publication in Materials, but it can be considered for publication after the authors make major revisions as follows.

  1. I would encourage and advise you to adopt some of the additional references published by MDPI in the introduction section:

Tresca Stress Simulation of Metal-on-Metal Total Hip Arthroplasty during Normal Walking Activity. Materials (Basel). 2021, 14, 7554. https://doi.org/10.3390/ma14247554

The Effect of Bottom Profile Dimples on the Femoral Head on Wear in Metal-on-Metal Total Hip Arthroplasty. J. Funct. Biomater. 2021, 12, 38. https://doi.org/10.3390/jfb12020038

  1. Describe the novelty of the research article made by the author? From the results of my evaluation, it seems that there are many similar published works that adequately explain what you have raised in the current manuscript. This is the fundamental reason why this manuscript needs a major revision or is rejected.
  2. The flow of writing made by the author in making this article less “flowing” so it is quite difficult to understand and not interesting. This is the fundamental reason why this manuscript needs a major revision or is rejected.
  3. The author must provide an image as an illustration in the Materials and Methods section so that the reader can easily understand your research because what the author presents is only in the form of words. This is the fundamental reason why this manuscript needs a major revision or is rejected.
  4. Some of the paragraphs made by the author do not provide explanations that are less systematic and unclear because they only consist of one sentence, such as the example on lines 337-340. This is the fundamental reason why this manuscript needs a major revision or is rejected.
  5. The author must provide a brief specification regarding the tools used in the research carried out so that the reader can estimate the accuracy and differences in the results that the authors describe due to the use of different tools in future studies. For example, energy-dispersive spectroscopy (EDS) (VANTAGE Digital Microanalysis System, Noran Instruments Inc., Middleton, USA) in lines 133-135.
  6. In the results and discussion section, the author must describe the limitation in the research carried out, in the current article the author does not provide this.
  7. The conclusion of this manuscript is not solid. Further elaboration is needed. Further research also needs to be added in the conclusion.

Author Response

The authors wish to thank the reviewers for the detailed comments they held on our manuscript. Please find enclosed the point-by-point responses to the Reviewers’ comments. The changes made in the text were highlighted in yellow color for Reviewer 1, and green color for Reviewer 2.

# Reviewer 2:

1) Based on the results of an in-depth evaluation that I have done for a research article with the title "Mapping bone marrow cell response from senile female rats on 1 Ca-P-doped titanium coating", I think this manuscript should be rejected because it is not appropriate for publication in Materials, but it can be considered for publication after the authors make major revisions as follows.

Response: Thank you for your consideration in a possible publication after revision. We really appreciate that. About your suggestions, we have done a delicate revision adding some references, discussing more the innovation and differences of our study compared to other ones, and showing the reason this in vitro study should be considered for Materials Journal. All aspects were detailed answered, according to the listed below:

2) I would encourage and advise you to adopt some of the additional references published by MDPI in the introduction section:

- Tresca Stress Simulation of Metal-on-Metal Total Hip Arthroplasty during Normal Walking Activity. Materials (Basel). 2021, 14, 7554. https://doi.org/10.3390/ma14247554

-The Effect of Bottom Profile Dimples on the Femoral Head on Wear in Metal-on-Metal Total Hip Arthroplasty. J. Funct. Biomater. 2021, 12, 38. https://doi.org/10.3390/jfb12020038

Response: Thank you for your suggestion. In fact, such suggested MDPI references can add to the content of the introduction. The articles suggested were added to the article.

3) Describe the novelty of the research article made by the author? From the results of my evaluation, it seems that there are many similar published works that adequately explain what you have raised in the current manuscript. This is the fundamental reason why this manuscript needs a major revision or is rejected.

Response: Thank you for your suggestion. We completely agree with the reviewer. The novelty of this research proposal is essential for the study's justification. Thus, in the introduction section, a paragraph was added (page 5; second paragraph).

4) The flow of writing made by the author in making this article less “flowing” so it is quite difficult to understand and not interesting. This is the fundamental reason why this manuscript needs a major revision or is rejected.

Response: Thank you for your consideration. The revision was performed with the aim of improving the reading of the text, making the reading dynamic and clearer. Besides that, it is important to emphasize that the text was revised by the co-authors, some co-authors are fluent in the English language and one of them is an American university professor.

5) The author must provide an image as an illustration in the Materials and Methods section so that the reader can easily understand your research because what the author presents is only in the form of words. This is the fundamental reason why this manuscript needs a major revision or is rejected.

Response: A figure showing the experimental design, including the timeline, was added (all figures were re-organized to follow the right sequence of numbers). The new figure can be found in the text (page 9) and as follow:

Figure 1: Schematic representation of the experimental groups and periods of the analyses.

6) Some of the paragraphs made by the author do not provide explanations that are less systematic and unclear because they only consist of one sentence, such as the example on lines 337-340. This is the fundamental reason why this manuscript needs a major revision or is rejected.

Response: Sorry about that inconvenience. All of the text was revised in order to become clearer and consistent. The changes were highlighted in green color.

7) The author must provide a brief specification regarding the tools used in the research carried out so that the reader can estimate the accuracy and differences in the results that the authors describe due to the use of different tools in future studies. For example, energy-dispersive spectroscopy (EDS) (VANTAGE Digital Microanalysis System, Noran Instruments Inc., Middleton, USA) in lines 133-135.

Response: Thank you for your suggestion. It was done.

8) In the results and discussion section, the author must describe the limitation in the research carried out, in the current article the author does not provide this.

Response: Thank you for your suggestion. The limitations of our study were added in the results and discussion section (on page 17, last paragraph, highlighted in green color).

9) The conclusion of this manuscript is not solid. Further elaboration is needed. Further research also needs to be added in the conclusion.

Response: We agree with the reviewer. The conclusion was re-written adding the information that future clinical research is necessary (highlighted in green color).

Round 2

Reviewer 2 Report

Comments are sufficient. Thanks.

This manuscript is a resubmission of an earlier submission. The following is a list of the peer review reports and author responses from that submission.

Round 1

Reviewer 1 Report

Review of the manuscript "Mapping bone marrow cell response from senile female rats on Ca-P-doped titanium coating". The manuscript presents the results of a comparison of two surface treatment methods PEO and SLA. The main problems of the manuscript are the lack of controls and the abundance of relative measurements.
1. Lack of control experiments. The manuscript only compares the performance of PEO and SLA surface treatment methods. In classical evidence-based science, there are at least two components: control (rough workpiece) and experience (finished workpiece). There is often positive control (in this case, commercial SLA technology). There is no classical control in this manuscript, we have nothing to compare the results with.
2. Introduction. It remains unclear from the introduction why the comparison between PEO and SLA surface treatment methods at all remains. Also, the choice of alloy for surface modification was not clear to me. What is the beauty of Ti-6Al-4V alloy? Why was this particular alloy chosen?
3. In fig. 1B presents very interesting data. Oxygen 46.8% by weight. As we know, all carbon oxides are gases and cannot be found on the surface. Higher vanadium oxide - V2O5 (vanadium weighs 50, oxygen weighs 16). That is, 12.9% of the vanadium mass accounts for 10.3% oxygen. Titanium has the highest oxide TiO2 (titanium - 47, two oxygen 32), that is, titanium oxide can account for a maximum of 7.3% oxygen. Calcium has oxide CaO (calcium - 40, oxygen - 16), that is, oxygen is 3.5%. Thus, oxides account for about 20% of the mass of oxygen, let another 5% fall on the oxides of impurities. From this it follows that 20% of the mass of oxygen is not found anywhere or is on the surface in a liquefied state.
4. Cell viability is measured spectrally at 570 nm. What is the meaning of such measurements? Obviously, the cells were detached from the surface at each measured point. That is, the graph is assembled from many qualitatively related experiments?
5. Why were all measurements taken at 3,7 or 10 days, and mineralization at 21 days? On what day was alkaline phosphatase activity measured?

Minor Point. In paragraph 4.7. Cell Growth and Viability is written to measure at 170 nm. It should be noted that 170 nm is absorbed even in the spectrometer path (unless the path is vacuum). Perhaps the authors were wrong? 

Reviewer 2 Report

The article's intention is interesting to study the influence of the surface of titanium implants and the quality of the tissue (lowgrade bone) in the osseointegration processes; however, the experiments carried out, and the results are not enough and lack depth.

There are some writing-English problems in the text, for instance:                

"Biomaterials need to have osteoinduction, osteoconduction, and osteopromotion properties it should be Biomaterials for bone replacement

"Roughness and porosity are two interdependent surface characteristics of bioactive surfaces that result from the manufacturing process used."

Please confirm the concept of bioactive materials.

"Thus, given the possibility to create bioactive coatings on implant surfaces is gaining notoriety. Among them, a new method that has been investigated, electrolytic plasma oxidation (PEO), allows creating through micro discharges, a coating with high adhesive strength and micropores with the ability to incorporate ions (Ca2+ , P5+ , Zn2+ , Mg2+ , Ag1+ , Sr2+ , Si2+) [6–16]. This coating is similar to bioactive ceramics formed from PEO, also known as MAO (Micro arc Oxidation), an advanced electrochemical technique based on anodizing at high voltages exceeding the dielectric breakdown voltage of the oxide layer on the Ti surface and the gas envelope [6–10,13–18]. "Redundant information. Mentioning PEO two times in a confusing way

"However, no consolidated information can yet be found in the literature about its bone cell behavior in a lowgrade bone." Is this true???

"O PEO coating was conducted through an electrical system and a reactor consisting of an electrode holder and electrolytic tank. "What is an O PEO?

O PEO?

"Chemical characterization One representative specimen from each group was analyzed by elemental chemical analyses were performed in small volumes, of the order of 1 μm3 , using energy-dispersive spectroscopy (EDS) (VANTAGE Digital Microanalysis System, Noran Instruments Inc., Middleton, USA). Using Image J software (National Institutes of Health, USA), the average pore size and composite diameter deposited for each type of surface treatment were determined [19]." Check English

In the title and the introduction, the recipient bone quality's effect in the osseointegration process is discussed, but experiments, results, discussion, and conclusion did not clarify this topic. In the experimental part, although cells are obtained from adult animals, it would have been interesting to compare with cells from young and healthy animals.

The characterization of the material is shallow and low. Very little information on the description of the sample is shown. In Fig.1, the colors cannot be differentiated, and it is confusing to know which corresponds to which element. It would be essential to see images of the surface, morphology, the thickness of the coating, and EDS (mapping) of the sample.

Some inconsistencies are observed, such as:

It is mentioned that 50 samples were used (25 for one treatment and 25 for the other). When counting all the samples, they would be proliferation: 4 for each time Total:12. PCR 12 samples. Alkaline phosphatase ???. Mineralization 5. There are more than 25 samples for each group of surface treatment.

For Fig. 2 about cell proliferation, where are the results of the control cells? Were these values normalized?
The slope of the two curves (trend) is similar even though the starting cell population differs.

Why were selected 3, 7, and 10 days for the proliferation test? Why, for the other experiments, only 3 or 7 days were considered?

The text said that in Fig.4 the PEO-treated surface showed a higher mineralized bone formation (p = 0.002) in a mineralized matrix formation, but this is not reflected in the graph.

The conclusion is very general and not related to the hypothesis that was proposed:

"Obtaining a new surface treatment, which presents a similarity or even a superiority in cell differentiation and mineralization, already established by the SLA surface treatment method, as observed in this study, added with the possibility of adhesion of ions and molecules, a better wear and corrosion resistance, and better thermal protection [9,13,17,18,56,57], shows itself as a promising future for this surface treatment."

Is it a new surface treatment? Coatings with Ca and P were reported before using PEO. No conclusion was mentioned about the relationship between surface modification and the quality of the bone.